# Exerkines, Nutrition, and Systemic Metabolism

**DOI:** 10.3390/nu16030410

**Published:** 2024-01-30

**Authors:** Bruce A. Watkins, Brenda J. Smith, Stella Lucia Volpe, Chwan-Li Shen

**Affiliations:** 1Department of Nutrition, University of California, Davis, CA 95616, USA; 2Department of Obstetrics and Gynecology, School of Medicine, Indiana University, Indianapolis, IN 46202, USA; bsm14@iu.edu; 3Indiana Center for Musculoskeletal Health, School of Medicine, Indiana University, Indianapolis, IN 46202, USA; 4Department of Human Nutrition, Foods, and Exercise, Virginia Polytechnic Institute and State University (Virginia Tech), Blacksburg, VA 24061, USA; stellalv@vt.edu; 5Department of Pathology, Texas Tech University Health Sciences Center, Lubbock, TX 79430, USA; leslie.shen@ttuhsc.edu; 6Center of Excellence for Integrative Health, Texas Tech University Health Sciences Center, Lubbock, TX 79430, USA; 7Center of Excellence for Translational Neuroscience and Therapeutics, Texas Tech University Health Sciences Center, Lubbock, TX 79430, USA

**Keywords:** exerkines, metabolites, metabolomics, systemic metabolism, endocannabinoids, oxylipins, neuroinflammation

## Abstract

The cornerstones of good health are exercise, proper food, and sound nutrition. Physical exercise should be a lifelong routine, supported by proper food selections to satisfy nutrient requirements based on energy needs, energy management, and variety to achieve optimal metabolism and physiology. The human body is sustained by intermediary and systemic metabolism integrating the physiologic processes for cells, tissues, organs, and systems. Recently, interest in specific metabolites, growth factors, cytokines, and hormones called exerkines has emerged to explain cooperation between nutrient supply organs and the brain during exercise. Exerkines consist of different compounds described as signaling moiety released during and after exercise. Examples of exerkines include oxylipin 12, 13 diHOME, lipid hormone adiponectin, growth factor BDNF, metabolite lactate, reactive oxygen species (ROS), including products of fatty acid oxidation, and cytokines such as interleukin-6. At this point, it is believed that exerkines are immediate, fast, and long-lasting factors resulting from exercise to support body energy needs with an emphasis on the brain. Although exerkines that are directly a product of macronutrient metabolism such as lactate, and result from catabolism is not surprising. Furthermore, other metabolites of macronutrient metabolism seem to be candidate exerkines. The exerkines originate from muscle, adipose, and liver and support brain metabolism, energy, and physiology. The purpose of this review is to integrate the actions of exerkines with respect to metabolism that occurs during exercise and propose other participating factors of exercise and brain physiology. The role of diet and macronutrients that influence metabolism and, consequently, the impact of exercise will be discussed. This review will also describe the evidence for PUFA, their metabolic and physiologic derivatives endocannabinoids, and oxylipins that validate them being exerkines. The intent is to present additional insights to better understand exerkines with respect to systemic metabolism.

## 1. Introduction

Exercise in various forms of physical activity supports fitness, improves quality of life, and helps delay the onset of diet-related chronic diseases. In 2016, Safdar et al. [1] first used the term “exerkines” to describe compounds or signaling molecules (autocrine, paracrine, or endocrine processes) released in response to exercise that stimulate crosstalk between cells and within and across tissues, organs, and systems as a form of inter-organ communication with the CNS [2]. Exercise is known to have both acute and chronic effects on tissues, which can result from exercise-induced changes in gene expression [3]. However, exercise also has acute actions on metabolism and long-term actions or training effects resulting from adaptations that are driven by alterations in mRNA and protein expression of enzymes, receptors, and growth factors. In this context, exercise actions on whole-body metabolism and physiology, where exerkines are involved, are best understood from the current knowledge of energy catabolism and balance. Thus, it is important to study exerkines beyond the conventional knowledge of metabolites, hormones, and cell signaling factors. Herein, the focus on exerkines is to discuss their effects in relationship to the triad of macronutrient metabolism, diet, and physiology, especially as it relates to the central nervous system (CNS).

Exerkines are reported to afford the benefits of exercise that impact metabolism and the CNS. Exerkines influence the nervous system by improving nerve regeneration and synaptic plasticity [4], not surprisingly in conjunction with brain-derived neurotrophic factor (BDNF), and the endocannabinoids (eCB) that influence the effects of BDNF in the brain [5]. Exercise actions improve cognitive functions via adult hippocampal neurogenesis, dendritic remodeling, and synaptic plasticity [6]. Although BDNF is now considered to be an exerkine [6], exercise actions support neurogenesis, neuroplasticity, and BDNF concentrations that appear to involve the eCB [5]. BDNF expression has been reported to be higher in rodents (brain and serum) and humans (plasma and serum) after exercise [4]. The experimental findings suggest that docosahexaenoic acid (DHA) and related ethanolamide (*N*-docosahexaenoyl ethanolamide (DHEA)), an eCB-like compound, exert some control on BDNF, and exercise increases eCB *N*-arachidonoyl ethanolamide, or anandamide (AEA), which is an *N*-acylethanolamine [5]. The eCB, AEA, and 2-arachidonoylglycerol (2-AG) belonging to this well-recognized group of lipid-derived eCB compounds are produced during exercise and support brain physiology; thus, collectively, the eCB would seem to be likely candidate exerkines because they exert actions on the brain and intermediary metabolism [5].

The emphasis of this review is on exerkines associated with metabolism and the brain. Particular attention is given to other metabolites and the family of bioactive lipids, eCB, that influence systemic metabolism and brain physiology during and after physical exercise.

## 2. Overview of Exerkines Associated with Macronutrient Metabolism

One of the functions of exerkines is to support systemic and intermediary metabolism between supply organs to sustain energy needs and physiological processes of the brain during exercise. The metabolites from catabolic pathways of metabolism generated by energy supply organs during exercise are shown in Figure 1. The term exerkine has been defined as a collective group of messengers released during exercise, from major organs of nutrient metabolism, that impact the brain and peripheral nervous system [3]. Not surprisingly, the collective hormones, cytokines, and metabolites, now referred to as exerkines, work in concert to support all body functions during exercise beyond the brain. It is a new terminology, exerkines for compounds that mediate the actions of exercise, and now perceived as exercise communicators to sustain the CNS. However, most exerkines are of known origin and function but are now characterized and believed to support the positive results of the exercise to sustain the metabolic and physiological requirements of the brain and the entire body.

### 2.1. Classification of Exerkines Based on Their Source and Actions

To begin to understand this broad category of compounds and/or signaling molecules that are released in response to exercise, it is best to develop a systematic approach. For example, some exerkines are metabolic intermediates and could be classified as *metabolic exerkines* based on the metabolic pathways of macronutrients influenced by daily intakes, metabolic intermediates, and catabolic end-products. Still, other exerkines that include growth factors, cell signaling molecules, and hormones could be classified as a different group named *physiologic exerkines* based on their chemical form and target actions (tissues, cells, and subcellular organelles). This approach better characterizes the collective group of compounds by source and actions (metabolic and physiologic) within the two groups. The benefit of using a systematic approach for investigations into exerkines is to link their functions and collective actions in target organs, tissues, and cells. Therefore, metabolic exerkines and physiologic exerkines would be a logical classification to best understand how these compounds serve the body as a harmonious system where all parts cooperate for the functional whole.

Exerkines represent a paradigm to describe and integrate the effects of exercise on systemic metabolism for homeostasis and caloric balance for energy needs. Further, we perceive exerkine roles in the shifting impact of the degree of exercise, food selections, and acquiring proper nutrition on systemic metabolism as the body adapts to our energy needs (positive, negative, or neutral energy balance) and physiologic demands. The impact of exerkines is dependent on the type of exercise (intensity, duration, frequency, and modality) or lack thereof because of the dynamic shifts in metabolic pathways based on macronutrient substrates and catabolic metabolism. The exerkine’s integrative actions in metabolism and physiology serve to sustain the body with oversight by the nervous and endocrine systems during exercise.

### 2.2. Exerkines of Metabolism and Energy Production

One characterized metabolic exerkine is lactate, which is a product of glycolysis and the breakdown of glycogen as a result of glycogenolysis (Figure 2 Panel A). The catabolism of glucose yields pyruvate, which can be converted to lactic acid or form acetyl-CoA in the mitochondria to condense with oxaloacetic acid to yield citate in the TCA cycle (Figure 2 Panel A). Under physical exercise, muscle produces lactate continuously and is released into the blood. The liver takes lactate and, via gluconeogenesis, produces glucose a Cori cycle process where extrahepatic tissues oxidize glucose to yield lactate via anaerobic glycolysis.

Interestingly, intermediates of the TCA cycle may leak from mitochondria and have immunological actions that influence immune cells and, once activated, may alter the metabolic flux of the TCA cycle described as the “immunologic Warburg effect” [7]. Under certain exercise activities that lead to stress, the TCA intermediates can activate or suppress the immune system’s T lymphocytes [8]. Thus, understanding the TCA cycle under exercise conditions might lead to other metabolites that function as exerkines acting on immune cells producing cytokines to influence inflammation.

If pyruvate proceeds to acetyl-CoA, the potential to form glucose from pyruvate is lost (Figure 2 Panel A). Pyruvate can cross the blood–brain barrier and is an excellent substrate for cerebral energy metabolism and neurons [9]. Based on how the brain uses pyruvate and its availability during increased flux through glycolysis, it is conceivable that pyruvate is a candidate for exerkine. As pyruvate dehydrogenase (PDH) is upregulated, acetyl-CoA is directed into the TCA cycle for energy production. The brain, kidney, and heart have highly active forms of PDH in the fed state. However, during exercise, PDH activation in human skeletal muscle is proportional to the relative aerobic power output (percent VO_2_max) and is regulated by increases in Ca^2+^, free ADP, and pyruvate concentrations [10].

An important biosynthetic pathway to support the metabolism of all macronutrients (glucose, fatty acids, and amino acids) is the formation of coenzyme A (Figure 2 Panel B), which is required for anabolic and catabolic pathways. The formation of activated substrates is fundamental to support all macronutrient use as energy depending on the type and degree of exercise and the flux of metabolic pathways to produce ATP.

Exercise does induce the anabolic pathway gluconeogenesis via the alanine glucose cycle shown in Figure 3. During exercise alanine, a gluconeogenic amino acid can be used to produce glucose in the liver (Figure 3). The generation of glucose is an important part of metabolism to maintain blood glucose for the brain and red blood cells during exercise.

Exerkine actions on metabolism serve to support the following during exercise: first, the biochemical/metabolic consequences of macronutrient catabolism and anabolism for organs, tissues, and cells to supply reducing equivalents for electron transport to generate ATP; second, the physiology of muscle contraction and the engaged pathways of metabolism that maintain energy levels from glucose and fatty acid catabolism to sustain the body during exercise when energy expenditure is high. Metabolic flux through biochemical pathways is influenced by the types of substrates glucose, fatty acids, and amino acids used in exercise. For example, if glucose is plentiful, more energy is derived from glycolysis, but if lipolysis is engaged, more fatty acids are oxidized, and acetyl CoA is used in the TCA cycle. Therefore, the movement or flux of substrates and metabolites increases in a metabolic pathway to support exercise. In this scenario, organ functions during exercise and pathways related to systemic and intermediary metabolism are activated to maintain the high energy expenditure of the body as a form of inter-organ communication [2].

Muscle activity during exercise requires considerable energy from metabolic pathways. However, muscle contraction releases cytokines, e.g., various interleukins like IL-6 (a myokine) [3], and metabolite lactates derived from pyruvate during glucose catabolism. The relation between metabolic and physiologic exerkines must be mentioned at this point because of the effects in the muscle on energy production. Interleukins and growth factors exert autocrine and paracrine effects on muscles as exerkines [3]. Interestingly, IL-6, lactate, and adiponectin have supportive actions on brain mitochondrial function [11]. Adiponectin and BDNF are both proposed exerkines, adipose is a primary source of adiponectin [12], BDNF originates from immune cells [13], and both act in an autocrine fashion [3]. In muscles, adiponectin supports glucose use and fatty acid oxidation, and this adipokine increases with strenuous exercise [12]. Thus, in some cases, physiologic exerkines can influence metabolic exerkines.

Undoubtedly, understanding the complexity of exercise actions on systemic metabolism and physiology by identifying all potential exerkines and their interplay is key to gaining insights into the actions of exerkines on health, such as improved well-being, cognition, and psychological state. One experimental approach will be to characterize the metabolomic and genomic characteristics for the actions of groups of exerkines during exercise and translation to health benefits. Another research strategy is to identify the extent of exerkines’ involvement at all levels of physical activity and in health for all phases of the life cycle [14]. For example, Tai Chi, a mind–body exercise, not only changes the physical attributes of individuals but also controls the perception of pain [15] and changes the blood concentrations of eCB and oxylipins (OxL) that contribute to inflammation [16]. Metabolite-derived exerkines might be linked to different metabolic set points in intermediary metabolism and nutrient control, such as the metabolomic study on glucose use and changes in metabolic pathways for glucose use [17].

Different types of exerkines may be more important during growth, maintenance, and aging [14], and identifying their actions may best support the desired outcome for fitness and health, especially with respect to oxidative stress and inflammation that contribute to chronic diseases [3]. Physical attributes and biomarkers have been proposed [18] to reflect the benefits of exercise on cardiovascular and metabolic diseases. However, the focus on exerkines throughout the life cycle has yet to be considered. Other investigators suggest that exercise performance must incorporate markers of nutrition and metabolic status, hydration, muscle fitness, endurance, injury risk, and state of inflammation [19]. All these issues must be investigated to identify where exerkines are important and where their concerted actions occur systemically, as well as in the brain.

### 2.3. Hormones, Autocrine and Paracrine Exerkines, and Exercise Effects on Metabolism

The collective functions of exerkines are to dispatch endocrine, paracrine, and autocrine effects and supply nutrients directed toward the brain [3]. The target for exerkine actions during exercise includes benefits on the aging brain, e.g., enhanced neurogenesis and neuronal differentiation, increased BDNF concentrations, and improved spatial learning and memory, and many of these effects can be transferred through the administration of BDNF [20]. For these effects, the evidence is strong for BDNF to be grouped with the exerkines.

Many challenges arise in understanding exercise effects on exerkine concentrations and the brain; one is the differences due to exercise focus between women and men [18]. For example, differences in body composition and hormones, dietary intake, and the type of exercise women may gravitate to compared to men. There may be differences in activity, such as regular walking or recreational activities, whereas some prefer strengthening exercise or competitive sports. In addition, body composition, blood biomarkers, and dietary intake are other factors. Differences in the preference and motivation for exercising between women and men in this study may be significant mediators and network structures based on differential correlations between the exercise and non-exercise groups [18]. Recognizing variations in food preferences and dietary macronutrient intakes between women and men are many, and the impact is reflected in differences in metabolic flux through pathways (Figure 2 Panel A) that must be considered in understanding the control of these interrelated pathways during exercise [21]. Of course, individual preferences and goals are not only due to sex but are related to the type of exercise, training, diet, and age.

Exercise actions on metabolic pathways result in an elevation in metabolite concentrations of supporting organs illustrated in Figure 1. Changes in the concentrations of metabolites are due to how the body adapts from endurance training to influence glucose oxidation, TCA cycle metabolites, and electron transport system activity, which metabolites increase. Under this physiological condition, metabolic flux is increased in pathways, as is fatty acid transport from adipose, and muscle triglyceride breakdown, liberating fatty acids [22] that are activated and transported through the mitochondrial membrane by carnitine [23]. Fatty acids are combined with acyl-CoA for activation and enter beta-oxidation to yield acetyl-CoA, which enters the TCA cycle. It is well recognized that both glucose and fatty acids are oxidized during exercise, but the utilization of both has a limiting effect on glucose use to spare glucose for RBC that lacks mitochondria. The case for lactate as an exerkine (myokine) is closely linked to the brain and systemic energy use but exceeds in a much greater systemic purpose in tissues and organs [24]. Lactate has a sparing effect on glucose, which is essential for RBC, and lactate induces mitochondria in the brain to derive more ATP from the electron transport chain. Lactate increases the amounts of reducing equivalences and the utilization of more acetyl-CoA, resulting in greater flux through the TCA [25]. Exercise increases arterial lactate concentration and blood flow to enhance brain lactate supply; the brain takes up lactate predominately produced by the muscles as a substitute for glucose [24]. During heavy exercise, cerebral lactate uptake is important to neurons and BDNF secretion in the hippocampus [24]. In addition, lactate is fundamental for mitochondrial biogenesis in the brain [24].

Another potential exerkine to consider is pyruvate because exercise can elevate the level of this metabolite in tissues and blood. Supporting this idea is the protective effect of pyruvate on neurons in the brain [26]. Pyruvate can cross the blood–brain barrier and be converted to lactate [27], pyruvate is a fundamental metabolite for glucose synthesis (gluconeogenesis), and pyruvate is a carbon substrate for the TCA cycle, as shown in Figure 2 Panel A.

Understanding macronutrient used in metabolic pathways for acute exercise and adaptations to long-term training where exerkines participate must be elucidated. The specific role of the exerkines should also be investigated for cardiovascular health during physical exercise [28]. These aspects of metabolism and health further emphasize justification for studying metabolic exerkines versus physiologic exerkines to best reveal the pleiotropic nature of the exercise.

## 3. Endocannabinoids, Oxylipins, and Polyunsaturated Fatty Acids in Exercise

Justification for eCB as exerkines is proposed based on their effects during and after exercise, as shown in Figure 4. Exercise such as running in humans results in an increase in eCB in the brain and blood [29,30,31]. The arachidonic acid-derived AEA and 2-AG concentrations were increased in 63 healthy participants after running at moderate intensity levels on a laboratory treadmill for 45 min; after that, the same participants walked for 45 min [30]. The researchers found that running and walking led to higher plasma concentrations of eCB and that participants felt increased euphoria [30]. This euphoric feeling in participants after exercise is now believed to be due to the release of neurochemicals, including eCB [29,32,33]. In addition, muscle response to exercise is linked to eCB effects on macronutrient metabolism [34], as described in Figure 4. For example, the endocannabinoid system (ECS) and, specifically, blocking the cannabinoid CB1 receptor resulted in increases in glucose/pyruvate metabolic enzymes and mitochondrial TCA cycle in mouse muscle [35]. Furthermore, in mice, both a high carbohydrate diet and CB1 receptor activity regulated seven key enzymes of the glycolytic pathway and TCA cycle [35]. These data show that the ECS, its ligands eCB, and receptors, are an integral part of brain physiology and metabolism in the muscle and brain.

An important aspect of eCB actions is their role in inflammation, specifically neuroinflammation [5]. During mind–body exercises such as Tai Chi, the physical assessment of pain was reduced in conjunction with lowered inflammatory mediators called OxL (especially PGE_2_), along with the eCB 1-arachidonoylglycerol and 2-arachidonoylglycerol (sum of 1,2-AG) and N-linoleylethanolamine (LEA) in women with knee osteoarthritic pain [15,16]. Incidentally, one OxL 12,13 diHOME is now included as a proposed exerkine with autocrine effects [3]. Furthermore, OxL responses to acute and chronic exercise, many of which increase, were recently reviewed [36]. A final point about eCB and OxL is that eCB shares a common aspect of OxL synthesis as a substrate for the cyclooxygenase (COX) enzyme and can lead to greater inflammatory mediators such as the prostanoid PGE_2_ [37].

Both AEA and 2-AG and their respective receptors (CB1 and CB2) have been associated with effects on anxiety and pain [31]. The ECS is also linked to inflammatory cytokines [5] and to COX expression [37]. Therefore, the eCB may influence inflammation directly or via the OxL, e.g., as a substrate for PGE_2_. Although the primary eCB AEA and 2-AG are derived from arachidonic acid the long-chain omega-3 polyunsaturated fatty acids (PUFA) docosahexaenoic acid (DHA) and eicosapentaenoic acid (EPA) can be converted to docosahexaenoyl ethanolamide (DHEA) and eicosapentaenoyl ethanolamide (EPEA), respectively, as endocannabinoid-like compounds [29]. The shifting PUFA sources, lowering arachidonic acid with DHA and EPA potentially reduces AEA and 2-AG and may lower inflammatory state [5]. The actions of exercise on the brain production of eCB and the modifying effects of dietary PUFA are illustrated in Figure 4. It is well established that AEA and 2-AG stimulate appetite. However, oleoylethanolamide (OEA) may inhibit hunger [38]. The physiological actions of eCB and related compounds would be important factors to consider for exercise and recovery.

## 4. Diet, Nutrients, and Exerkine Effects on Metabolism

Nutrients, macronutrient concentrations, and exercise influence the flux of metabolites through metabolic pathways based on the energy needs of tissues and organs, and systemic energy demands are dictated by physical exercise. Understanding flow through metabolic pathways and rate-limiting steps can be achieved with metabolic control analysis to appreciate the structural design and intricate inter-related metabolism of nutrients, metabolites, and body energy resources [21]. Furthermore, metabolic pathways are under the genetic control of rate-regulating enzymes and nutrient-dependent transcription factors [39]. As with some exerkines that have cell signaling properties, nutrients such as some fatty acids can activate the peroxisome proliferator-activated receptor (PPAR), and glucose-derived metabolites activate the carbohydrate response element-binding protein (ChREBP) [39]. Therefore, the activation of PPARs and ChREBP, which are nutrient-dependent transcription factors, are the control points of metabolic functions. Importantly, PPARα influences fatty acid oxidation, lipid and lipoprotein metabolism, and inflammation [40]. In addition, when considering nutrients that impact metabolism and how exercise influences energy expenditure are achieved by microRNAs, and many nutrients (Vitamin D and E, oleic acid, and sodium) control overall gene expressions and microRNA concentrations [41]. Interestingly, oleic acid is a precursor of OEA, which is an antagonist of the eCB anandamide and antiobesity factor that activates PPARᵧ [42]. Moreover, PUFA that binds to and activates PPARᵧ may be enhanced by other dietary factors, such as genistein, that increase mRNA for PPARᵧ [43]. Thus, another example of metabolic and physiologic control of fat oxidation and fat metabolism is mediated by an eCB-like compound OEA that blocks cannabinoid receptors for appetite.

In a cell culture study with mouse and primary human myoblasts, the level of glucose and lactic acid was higher with the treatment of DHEA and AEA compared to the BSA control [44], and the results are presented in Table 1. The potent and selective antagonist NESS0327 of cannabinoid receptor CB1 resulted in a similar increase in glucose and lactate in primary human myoblasts. Moreover, the phosphorylated, activated glucose intermediates and phosphoenolpyruvate were higher in these groups (Table 1). The responses of myoblasts to eCB and the antagonist of CB1 suggest that glucose catabolism is increased by DHA, DHEA, and AEA in humans but not mouse myoblasts. Pyruvate was only higher compared to the BSA control in the human myoblasts treated with DHA. Additionally, primary mouse myoblasts treated with DHA showed higher concentrations of DHA-containing lipids (ethanolamine) compared to the BSA control but not in human myoblasts (Table 1). The level of coenzyme A was higher in the DHA- and DHEA-treated human myoblasts compared to the BSA group but only in the DHA group of mouse primary myoblasts. The mouse myoblasts accumulate DHA in glycerol lipids but not human myoblasts (Table 1). These findings suggest that human myoblast metabolism of glucose is altered by eCB, and likely, DHA and DHEA compete for CB1 binding like the action of NESS0327. The metabolism of glucose is illustrated in Figure 2 Panel A.

Other studies in mouse C2C12 myoblast cultures showed that enriching cells with DHA and EPA increased the protein expressions of CB1 and CB2 in proliferating and differentiated cells compared to the BSA control [45]. DHA resulted in higher mRNA for both cannabinoid receptors in contrast to the actions of AEA; furthermore, glucose uptake was higher with the DHEA treatment of myoblasts [46]. In our confirmation of dietary DHA actions on mouse systemic metabolism, lower glucose, and lactate were observed from the metabolomic analysis of muscles and livers after feeding a semipurified diet [17]. The findings in myoblasts [45,46] and mice [17] are consistent with previous findings of CB1 activation studies on glycolytic pathway enzymes in mouse muscles [35].

In older women, dietary DHA and EPA altered serum eCB, OxL, and metabolites [47]. The participants demonstrated a shift in metabolites, and PUFA-related eCB and OxL shifted to a lower inflammatory state associated with the dietary supplementation of the *n*-3 PUFA, DHA, and EPA. A similar study showed that dialysis patients with lower concentrations of plasma EPA, compared to control participants, had higher concentrations of 2-AG and AEA but lower OxL (12-HEPE and 5-HEPE) derived from EPA [48]. Thus, diet specifically amounts and types of PUFA, which determine the specific derivative, eCB, alter actions on brain physiology during exercise. Furthermore, the changes in dietary PUFA influence the concentrations and types of OxL elaborated from exercise and consequently neuroinflammation.

Exercise improves brain insulin resistance and cognitive function in animal models and humans, which are associated with age-related decline in memory and cognition [49]. Therefore, diet and exercise will have an impact on exerkines such as lactate and modulate the types of eCB and OxL in the brain during and after exercise.

Global metabolomics was performed in a study on carbohydrate feeding to men to evaluate insulin sensitivity [50]. Serum profiling followed changes in men consuming varying daily carbohydrate intakes and showed a clear impact on the components of nutrient classes, including amino acids, fatty acids, and ketones (as determined by metabolomic analyses). The results revealed significant changes in serum lipids (Table 2). The serum of male participants (*n* = 9), age 44.9 ± 9.9 yr. and BMI 37.9 ± 6.3 kg/m^2^, was analyzed for metabolic concentrations after consuming a low (35 g), moderate (125 g), or high (350 g) daily intake of carbohydrates for three-week periods. A higher carbohydrate diet resulted in higher fatty acid concentrations, but differences were observed across diet groups (Table 2). The diets resulted in large changes in ketone production (acetoacetate and 3-hydroxybutyrate), as shown in Figure 5 panel A, catabolism of branched-chain amino acids (BCAA), as shown in Figure 5 panel B, and use of amino acid carbon in the TCA cycle (Figure 5 panel C). BCAA oxidation, which is higher under low CHO feeding to men, like valine, provides more carbon skeleton for gluconeogenesis. The change in metabolites reflects the macronutrient composition of the diets and their impact on metabolic pathway flux, which have implications on exercise demands as presented herein.

## 5. Exercise, Neurobiology, and Neuroinflammation

Exercise in both humans and animals is reported to support neuroplasticity and a lower risk of neurodegenerative disease [51,52]. Several lines of evidence link physical activity or structured exercise to multiple benefits for the brain, and the ECS is a modulator of factors that improve depression, anxiety, and mental illnesses [32]. Regarding exercise effects and the elaborated exerkines, the neurotrophin BDNF exerts protective actions on the CNS that are likely in combination with other exerkines [4,51]. Recently, the combined effects of BDNF and lactate, the latter increase BDNF, and the combination improves neurobiological physiology [52]. The positive effects of exercise on neuroplasticity, which is an underlying attribute to reducing dementia [52], appear to involve the combined actions of BDNF, eCB, and lactate.

Neuroinflammation is a condition where reactive oxygen species, cytokines, and other mediators induce inflammation in the brain and the peripheral nervous system [53]. Two groups of bioactive lipids and their derivations, eCB and OxL, influence inflammation in the brain [5], and the components of the ECS (ligands and receptors) can modulate brain physiology to impact memory, learning, stress, and emotion via brain plasticity [32].

At present, the eCB directly affects the brain and pain mechanisms, and the OxL participates in inflammatory processes of the central and peripheral nervous systems [5,16]. Exercise affords many benefits to the brain, and some types of exercise cause the release of eCB from the brain [30,31] and both eCB and OxL in the blood [16]. Understanding the actions of exercise on specific eCB and OxL is an approach to explore the full potential of exerkines and their impact on the brain. With respect to diet, PUFA of the *n*-3 and *n*-6 families is one non-invasive approach to alter the biosynthesis of eCB and OxL and diminish oxidative stress and inflammatory processes [54]. Reducing neuroinflammation is a means to reduce pain and improve health during recovery from exercise.

## 6. Exercise and the Brain

Exerkines in the brain support energy homeostasis [49], and the activation of the CB1 by eCB stimulates appetite [5]. Exercise promotes neurogenesis, increases the number of synapses between neurons (synaptogenesis), and improves blood vasculature and angiogenesis through growth factors released during exercise [49]. The association between physical activity and the ECS in studies shows strong evidence of positive attributes on brain physiology related to neurogenesis and synaptic plasticity [55]. However, the full extent of biological mechanisms where the ECS participates in exercise is not fully explained. Known relationships are shown in Figure 4.

It is widely recognized that exercise improves cognitive function and metabolic efficiency in the brain [56]. The benefits of lactate on brain health and executive function have been implicated in studies involving high-intensity interval exercise (HIIE) compared to moderate-intensity continuous exercise [57]. Higher levels of serum lactate are achieved with HIIE compared to moderate-intensity exercise. Furthermore, exercise and the exerkines have specific effects on the presynaptic and postsynaptic receptors of neurons [58]. From the metabolic perspective, astrocytes can remove glucose from circulation or degrade glycogen to liberate glucose and generate lactate [56]. Neurons can use lactate from the extracellular space that is generated from the catabolism of glucose via glycolysis (Figure 2). Lactate is an important substrate for energy production in the brain, supports synaptic activity [25], and acts as a signaling molecule [24]. Some evidence suggests that exercise influences the gene expression and activities of transporters and enzymes in the proposed astrocyte–neuron lactate shuttle and via the lactate receptor (hydroxycarboxylic acid receptor 2 HCARI) that modulates neuronal network activity and affects brain plasticity [56].

In a review of nineteen studies of physical exercise in adolescents (12–18 years of age), moderate aerobic and resistance plus aerobic exercises improved depression symptoms [59]. Studies indicate that depression is related to the dysfunction of neurotransmitters [59]. Both human and animal studies show that exercise increases mitochondrial functions in neurons, alters concentrations of neurotransmitters, increases neurotrophic factors, and lowers inflammatory mediators [59,60]. Exercise in young adults changes the concentrations of microRNA and also alters the expression of cardiac exercise testing and exercise training microRNAs in young male adults [61] and under different dietary habits [41].

Exerkines derived from skeletal muscle (lactate), liver, and adipose directly impact brain mitochondrial function [11]. Mitochondria serve an important capacity in energy production and the CNS regulation of energy use [11]. The specific actions of exerkines on mitochondrial bioenergetics translate to the neurogenesis, neuroplasticity, and cognitive function of exercise impacts on the brain [11]. Furthermore, regular exercise and exerkines elicit vital adaptations via redox signaling in the muscle, heart, liver, and brain [60].

Brain plasticity is the ability to adapt under changing conditions of learning experiences and requires that neurons alter the nature of connections in response to different stimuli [14]. Exercise and various forms of physical activity stimulate brain plasticity, which is mediated by increases in different growth factors, BDNF and eCB [14,30,31], and both eCB and OxL increase in the blood [16]. However, the exact mechanism for the ECS actions on brain plasticity and connectivity is not known. As the study of exercise impacts on brain neuroenergetics is unraveled, the field of exerkines will help explain the energetics of the CNS.

## 7. Conclusions and Future Perspectives

The purpose of this review was to integrate the actions of exerkines with systemic metabolism during exercise. Although lactate is an exerkine, other metabolites, such as pyruvate, would fall into this category of catabolic intermediates, considering the metabolic and physiologic functions of pyruvate during and after exercise. Based on its broad metabolic diversity, specific actions in the brain, and metabolic fate, there are reasons to include metabolites such as pyruvate as exerkines. New evidence that TCA cycle metabolites affect immune cells and inflammation could justify these as candidate exerkines. A goal of this review was to describe the evidence for dietary PUFA and their derivative bioactive lipids, eCB and OxL, as potential exerkines that are produced during exercise. The inclusion of these bioactive lipids would provide greater insights into the effects of exerkines on the systemic metabolism brain interface. Nevertheless, the evidence for the benefits of the ECS during exercise on the brain is overwhelming. Aspects of diet and its impact on metabolism and specific effects of PUFA were presented, where glucose use was altered in metabolism and in conjunction with changes in gene expressions of receptors for the ECS and glucose uptake. The types of eCB and OxL are also influenced by dietary PUFA, where *n*-3 PUFA resulted in more favorable responses to glucose use, which is important to support this exercise. Additionally, the level of carbohydrates fed to men significantly altered the intermediary metabolism of fatty acids, ketones, and amino acids, which has implications during exercise. Hence, dietary PUFA, as a substrate for eCB and OxL, which function as metabolic and physiologic factors in the brain and systemic metabolism, justify that these compounds should be considered exerkines.

Bearing in mind the diverse actions of exerkines as a future perspective, we proposed that two groups of exerkines be used to best understand and characterize their actions as metabolic exerkines and physiologic exerkines. This approach better characterizes the collective group of compounds by source and actions within the two groups. Future research on exercise and exerkines should be directed on how these compounds support the systemic metabolism of related organs during exercise to supply the brain.

## Figures and Tables

**Figure 1 nutrients-16-00410-f001:**
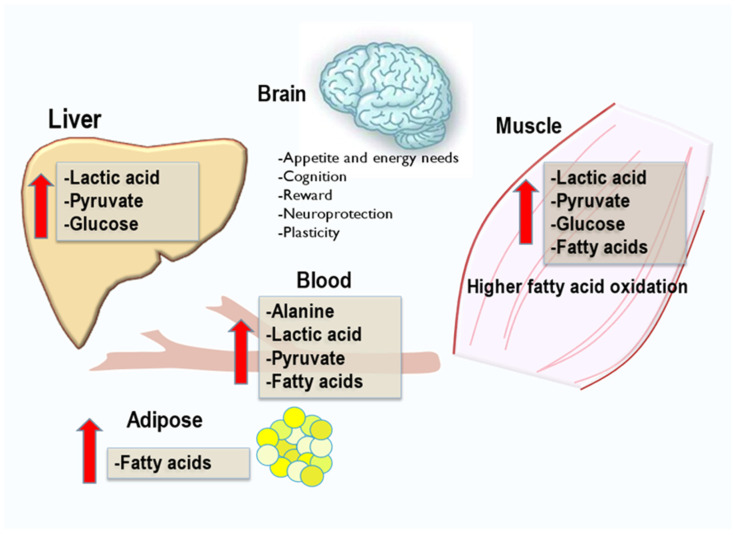
Overview of exercise effects on systemic metabolism in principal organs. Exercise induces many metabolic pathways for the generation of ATP for muscle contraction and other organ functions. Pathways of catabolism generate the production of lactic acid in muscle and liver and the production of fatty acids in adipose. Glycogen breakdown in muscle and liver releases glucose for muscle and brain; however, glucose is essential to support red blood cell functions to carry oxygen from the lungs in exchange for carbon dioxide produced by catabolism in tissues. Lactate is considered an exerkine.

**Figure 2 nutrients-16-00410-f002:**
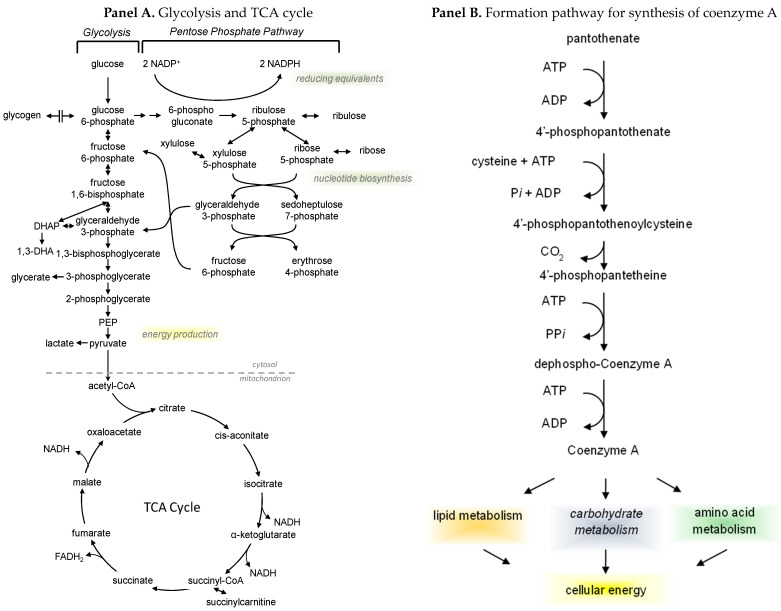
Integration of glycolysis with the tricarboxylic acid cycle (Panel A) and formation of coenzyme A (Panel B). Glycolysis and the tricarboxylic acid cycle (TCA cycle) are connected by pyruvate, which can proceed to lactate or acetyl-CoA, depending on the energy status of the cell (both occur in muscle and liver). However, once pyruvate is converted to acetyl-CoA, the carbon cannot be converted back to glucose. Pyruvate can be converted back to glucose in the liver. Glycolysis and the TCA cycle provide reducing equivalents for the production of ATP, the energy currency, for cell activity. The reducing equivalents are NADH and FADH_2_ for the electron transport chain in the production of ATP, and NADPH derived from the pentose phosphate pathway is required for fatty acid synthesis during the fed state such as with an abundance of glucose or amino acids, which are used in fatty acid synthesis. The pentose pathway is also important for nucleotide synthesis. The catabolism of fatty acids during exercise (beta-oxidation) yields acetyl-CoA that supplies substrate for the TCA cycle. Glycolysis resides in the cytoplasm and the TCA cycle in the mitochondria. Synthesis of acetyl-CoA is necessary for energy production to proceed during exercise from metabolic pathways for macronutrient substrates such as glucose, fatty acids, and amino acids. The mitochondria are the final steps for ATP production from reducing equivalences produced in glycolysis and the TCA cycle. Lactate from glycolysis is an important exerkine derived from glycolysis in muscle and brain. Synthesis of acetyl-CoA is necessary for energy production to proceed during exercise from metabolic pathways for macronutrient substrates such as glucose, fatty acids, and amino acids.

**Figure 3 nutrients-16-00410-f003:**
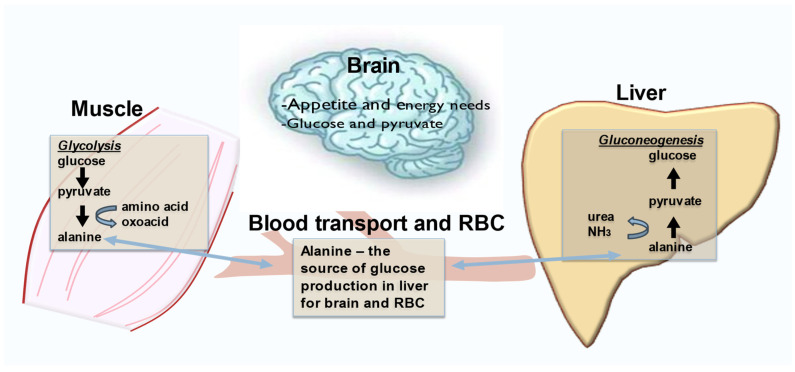
Exercise can induce the glucose alanine cycle. Depicted are the relationships between catabolism of the amino acid alanine in muscle during strenuous exercise to generate glucose needs for the brain and red blood cells (RBC). Thus, exercise actions can result in the conversion of pyruvate to alanine, a gluconeogenic amino acid that is transported from muscle via blood to the liver for gluconeogenesis.

**Figure 4 nutrients-16-00410-f004:**
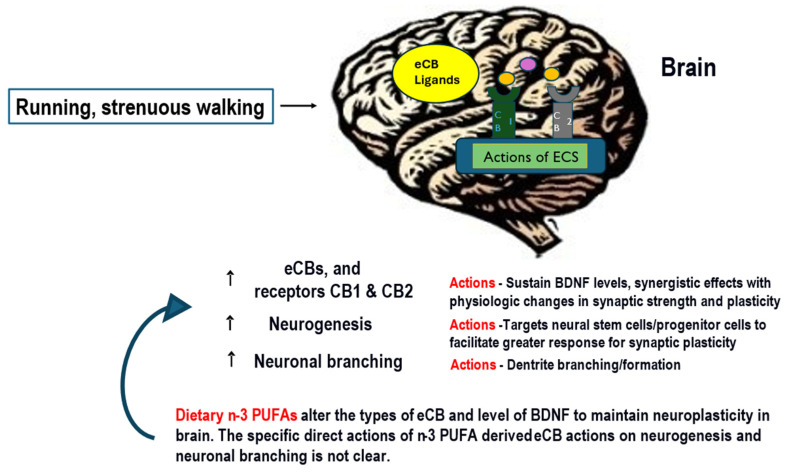
Exercise actions on the brain stimulate neurogenesis, neuroplasticity, higher brain expressions of BDNF, and levels of eCB in the brain and blood. Several types of exercise are known to increase eCB in the brain, and some increase BDNF. Exercise increases plasticity and neurogenesis and supports physiological changes that benefit the brain and improve well-being. The eCB are likely candidate exerkines based on their actions during exercise that includes increased metabolic flux of glycolysis and the TCA cycle. Exercise at a moderate level increases eCB. Dietary PUFA is the substrate for eCB synthesis. Increasing the dietary *n*-3 PUFA can lower arachidonic acid and the substrate for the synthesis of the arachidonic acid-derived eCB AEA and 1,2-AG. The eCB influences food intake and macronutrient metabolism (see text and references [33,34,35]). In the brain, eCB supports neuroplasticity. Abbreviations = polyunsaturated fatty acids = PUFA, eicosapentaenoic acid = EPA, docosahexaenoic acid = DHA, docosahexaenoyl ethanolamide = DHEA, eicosapentaenoyl ethanolamide = EPEA.

**Figure 5 nutrients-16-00410-f005:**
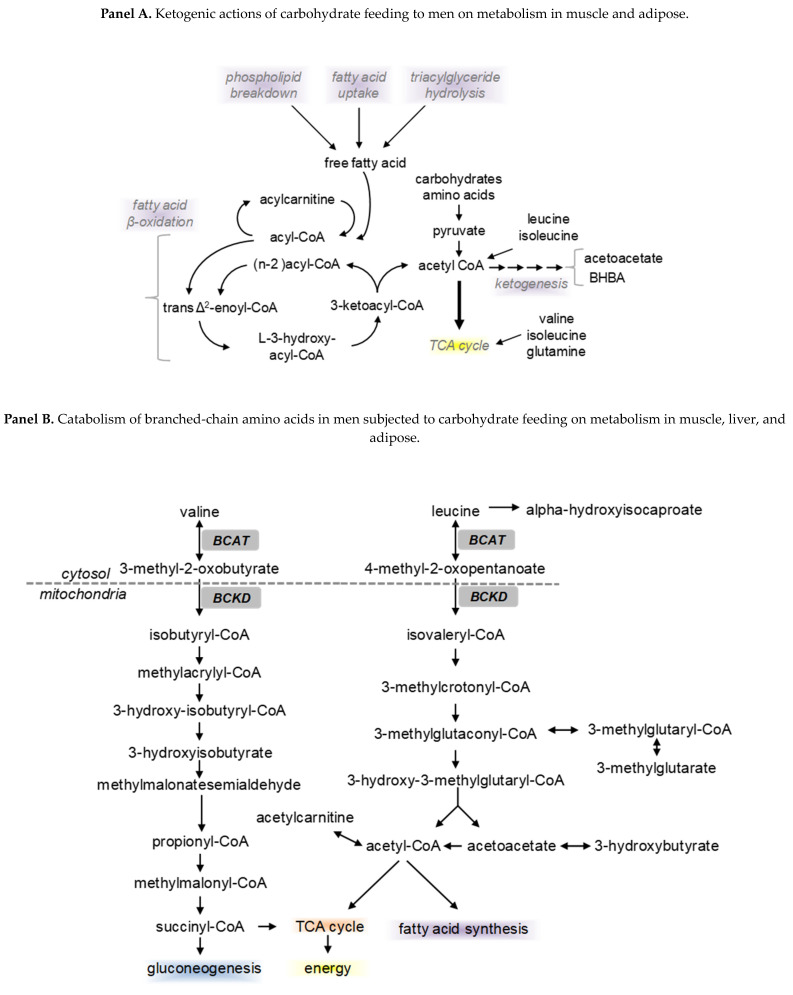
Effects of feeding different levels of dietary carbohydrate to men on metabolism. Serum metabolites in the same male subjects (*n* = 9) after consuming low (35 g), moderate (125 g), or high (350 g) daily intake of carbohydrates for 3-week periods (diets described in Table 2). Panel A illustrates how the low-carbohydrate diet induced a ketogenic response of lipid breakdown to liberate fatty acids for oxidation, ketogenesis, and the catabolism of amino acids to fuel the TCA cycle via acetyl CoA. Panel B represents the catabolism of branched-chain amino acids (BCAA) during a low carbohydrate diet. Panel C illustrates the catabolism of amino acids and their carbon skeleton to fuel the TCA cycle under a low carbohydrate diet (Watkins et al., 2014 [49]).

**Table 1 nutrients-16-00410-t001:** Effects of polyunsaturated fatty acids and endocannabinoids on oxylipins and metabolites in transformed, mouse, and human myoblasts. ↑ significantly higher, ↓ significantly lower, + approaching significance (higher), - approaching significance (lower).

Compound (Changes in Compounds, Treatments Compared to BSA)	C2C12					1° Mouse					1° Human				
	BSA	DHA	DHEA	AEA	NESS0327	BSA	DHA	DHEA	AEA	NESS0327	BSA	DHA	DHEA	AEA	NESS0327
docosahexaenoate		↑					↑					↑			
docosapentaenoate		↑					↑					↑			
docosahexaenoyl ethanolamide			↑												
6-keto prostaglandin F1alpha		↓					↓								
prostaglandin E2													+		↑
11-HETE		↓	-												
13-HODE + 9-HODE		↓	↓												
1-docosahexaenoylglycerophosphoethanolamine *		↑	↑				↑								
1-docosahexaenoylglycerophosphocholine (22:6n3) *							↑								
2-docosahexaenoylglycerophosphoethanolamine *		↑	↑				↑	+				+			
2-docosahexaenoylglycerophosphocholine *		↑	↑				↑							↓	
1-oleoylglycerophosphoethanolamine *												↓	↓		-
2-oleoylglycerophosphoethanolamine *												↓	↓	↓	
2-palmitoylglycerophosphoethanolamine *		↓	-									↓			
2-linoleoylglycerophosphoethanolamine *		↓	↓				-								
ethanolamine		+	↑	+											
choline			+												
glycerophosphorylcholine (GPC)												↑	↑		
choline phosphate			-		-		↓					↓	↑		
phosphoethanolamine			↑				↑					↑	↑		+
glycerol 3-phosphate (G3P)		↑		+			↑					↑	↑	+	↑
glycerol		↑	+												
1-docosahexaenoylglycerol (1-monodocosahexaenoin)		↑										↑			
1-palmitoylglycerol (1-monopalmitin)												+			
1-myristoylglycerol (1-monomyristin)		↑					+	+							
2-myristoylglycerol (2-monomyristin)		↑					↑	+							
1-linoleoylglycerol (1-monolinolein)		↑					↑	↑							
2-linoleoylglycerol (2-monolinolein)		↑					↑	↑							
glucose												↑	↑	↑	↑
glucose-6-phosphate (G6P)												↑	↑	↑	↑
fructose-6-phosphate (F6P)												↑	↑	↑	↑
3-phosphoglycerate													↑	↑	↑
2-phosphoglycerate													↑	↑	↑
phosphoenolpyruvate (PEP)													↑	↑	↑
pyruvate												↑	+	+	
nicotinamide adenine dinucleotide (NAD+)													+	↑	
nicotinamide adenine dinucleotide reduced (NADH)													↑	+	
lactate												+	↑	↑	↑
citrate												↓	-		
succinate													↑	+	↑
arginine													↑	↑	+
ornithine													+	↑	↑
putrescine												↑			
spermidine												↓	↓		
spermine												↓	-	↓	-
4-guanidinobutanoate												↑	-		
pantothenate		↓	+	↑								↓	↓		
phosphopantetheine		↑											+	↑	↑
3′-dephosphocoenzyme A							↑	↑				↑	↑	+	↑
coenzyme A							↑					↑	↑		

Myoblast cultures were treated with PUFA (DHA), endocannabinoids (DHEA and AEA), and antagonists (NESS037). All data are shown as the change from the BSA control myoblasts. Arrows indicate change relative to the BSA vehicle control (*p* ≤ 0.05). Metabolomic methods described previously [44,45,46]. Compounds with an “*” indicate differences in primary glycerolipids after DHA and DHEA treatments resulting in membrane remodeling of myoblasts.

**Table 2 nutrients-16-00410-t002:** Changes in serum fatty acids in male subjects given different levels of carbohydrates.

	Repeated Measures ANOVA Contrasts						Repeated Measures ANOVA Main Effect
Biochemical Name	35 g Baseline	125 g Baseline	350 g Baseline	125 g 35 g	350 g 35 g	350 g 125 g	
linoleate (18:2n6)	0.89	**0.8**	**0.72**	0.89	**0.81**	0.9	
linolenate [alpha or gamma; (18:3n3 or 6)]	** 0.77 **	**0.76**	**0.62**	0.99	0.81	0.81	
dihomo-linolenate (20:3n3 or n6)	**0.65**	**0.63**	** 0.78 **	0.97	1.2	** 1.24 **	
eicosapentaenoate (EPA; 20:5n3)	0.91	0.95	0.85	1.04	0.93	0.89	
docosapentaenoate (n3 DPA; 22:5n3)	0.8	0.81	0.81	1.02	1.02	1	
docosapentaenoate (n6 DPA; 22:5n6)	0.68	**0.66**	** 0.69 **	0.97	1.02	1.05	
docosahexaenoate (DHA; 22:6n3)	0.9	1.07	0.99	1.19	1.1	0.93	
myristate (14:0)	0.94	**0.8**	**0.68**	0.84	**0.73**	0.86	
myristoleate (14:1n5)	0.98	0.93	**0.7**	0.94	**0.71**	0.75	
pentadecanoate (15:0)	1.01	0.88	**0.71**	0.87	**0.7**	**0.81**	
palmitate (16:0)	1.09	**0.83**	**0.76**	** 0.77 **	**0.7**	0.91	
palmitoleate (16:1n7)	0.89	0.79	**0.59**	0.89	**0.66**	0.74	
margarate (17:0)	1.14	0.88	**0.79**	** 0.78 **	**0.7**	0.9	
10-heptadecenoate (17:1n7)	0.94	0.85	**0.69**	0.91	** 0.73 **	0.8	
stearate (18:0)	0.99	0.91	**0.75**	0.93	**0.76**	** 0.83 **	
oleate (18:1n9)	1.12	0.95	**0.77**	0.85	**0.69**	**0.81**	
cis-vaccenate (18:1n7)	1.04	0.92	**0.76**	0.88	**0.73**	** 0.83 **	
stearidonate (18:4n3)	**0.62**	**0.63**	**0.64**	1.02	1.04	1.03	
nonadecanoate (19:0)	1.18	1.01	**0.78**	0.86	**0.66**	**0.77**	
10-nonadecenoate (19:1n9)	** 1.23 **	0.92	**0.72**	**0.75**	**0.58**	** 0.78 **	
eicosenoate (20:1n9 or 11)	1.14	0.97	**0.7**	0.85	**0.61**	**0.72**	
dihomo-linoleate (20:2n6)	1.01	** 0.83 **	**0.75**	** 0.82 **	**0.75**	0.91	
arachidonate (20:4n6)	0.99	0.92	0.91	0.93	0.92	0.98	
adrenate (22:4n6)	0.89	** 0.74 **	0.8	0.83	0.89	1.07	

The data presented are a heat map of the statistical analysis of the metabolite measurements in the serum of men. Serum fatty acids metabolites in the same male subjects (*n* = 9) after consuming low (35 g), moderate (125 g), or high (350 g) daily intake of carbohydrates for 3-week periods [Watkins BA et al. 2014 [50]]. Values are presented as fold change from baseline for subjects. The diets consisted of beef, eggs, and dairy for daily protein sources throughout all diet phases as primary sources of saturated fat. For the low carbohydrate diet phases, higher-fat beef and meats, whole eggs, and full-fat dairy products (e.g., cheese, whole milk yogurt, cream, butter) were emphasized. For the higher carbohydrate diet phases with lower saturated fat, leaner versions of beef, egg substitutes, and low-fat dairy (e.g., reduced-fat dairy, skim milk, low-fat/non-fat yogurt) were used instead. Whole grain and relatively low glycemic index carbohydrate sources were emphasized. Heat map of statistically significant biochemicals profiled in this study. Red and green shaded cells indicate *p* ≤ 0.05 (red indicates that the mean values are significantly higher for that comparison; green values significantly lower). Light red and light green shaded cells indicate 0.05 < *p* < 0.10 (light red indicates that the mean values trend higher for that comparison; light green values trend lower). For the ANOVA, blue-shaded cells indicate *p* ≤ 0.05; light blue-shaded cells indicate 0.05 < *p* < 0.10.

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
