# Peer review of "Exerkines, Nutrition, and Systemic Metabolism"

_nutrients, 2024, doi:10.3390/nu16030410_

Round 1
Reviewer 1 Report
Comments and Suggestions for Authors
This review examined how various exerkines and diet alters metabolism, with a primary focus on lactate, BDNF and endocannabinoids (eCB). The main consideration of this review was that these exerkines alters CNS metabolism, decreases neuroinflammation, and have positive effects on brain physiology (neurogenesis and synaptic plasticity). Overall, the review is informative, but there is a fair amount of repetition, and the readability could be greatly improved with and more focused review, some editing/condensing, and a reduction (or combining of) in the number of figures. This review might be better if it focused on eCBs with exercise and diet/supplementation. Other, more specific comments are below:
Abstract: Several exerkines (e.g. IL-6 and oxylipins) are mentioned in the abstract, but a barely addressed in the review.
Sections 1.1 - 1.3: The composition of these section and the information in these sections is unorganized repetitive, and somewhat protracted. These sections should be combined and reduced.
Lines 76-89: This paragraph is repetitive regarding the number of times the two categories of exerkines are classified.
Lines 133-135: Also, consider the role that LDH has in converting lactate to pyruvate, which corresponds to the fact that blood lactate levels rise much more so than pyruvate during exercise.
Lines 149-151: This sentence is awkward and difficult to read.
Lines 168-185: This paragraph is somewhat repetitive with information earlier presented in the manuscript.
Lines 241-261: See comment regards Sections 1.1-1.3.
Figures and Tables: There are way too many figures, a few of them (e.g. figures 1&4, 5 &6) could be combined or are not necessary (e.g. figure 3).
Author Response
Abstract: Several exerkines (e.g., IL-6 and oxylipins) are mentioned in the abstract, but a barely addressed in the review.
Responses: The comment is recognized, and response carefully explained. One sentence described the variety of compounds called exerkines. However, as stated on line 57. “The purpose of this review is to integrate the actions of exerkines with respect to metabolism that occurs during exercise and propose other participating factors of exercise and brain physiology.
Sections 1.1 - 1.3: The composition of these section and the information in these sections is unorganized repetitive, and somewhat protracted. These sections should be combined and reduced.
Responses: The comments are acknowledged, responses thoughtfully considered, and revisions were made. The information presented builds on the rationale of concepts and are not protracted. The sections were revised following suggestions of this reviewer and some content deleted.
2.0 Overview of exerkines associated with macronutrient metabolism
2.1 Classification of exerkines based on their source and actions
2.2 Exerkines of metabolism and energy production2.3 Hormones, autocrine and paracrine exerkines, and exercise effects on metabolism
Many recent reviews have not taken the effort to distinguish between the specific metabolic and physiologic actions of exerkines thus it is necessary to differentiate these actions and integrate the biological impacts. We have done this in a reasonable, scientific context, and logically and concisely. The new sections delve into the science related to exerkines, metabolism, exercise, and physiology. Our premise is based on exerkine actions has not yet been proposed in the literature and is therefore a novel proposition of exercise science presented in our review. The presentation of the exerkine classification was done integrating organs, metabolites of pathways, and metabolite flux. All are important aspects for understanding exerkines. Further the use of different macronutrients (glucose and fatty acids) and related metabolic pathways engaged during exercise are discussed to support our idea. Lastly, we present some evidence about type of exercise and sex.
Lines 76-89: This paragraph is repetitive regarding the number of times the two categories of exerkines are classified.
Response: We disagree with the reviewer’s comment. These sentences are not repeating categories of exerkines. In contrast, these sentences serve to concisely explain what exerkines are and based on their functions as exerkines. The text is a logical description of what exerkines are and their role in metabolism and physiology.
Lines 133-135: Also, consider the role that LDH has in converting lactate to pyruvate, which corresponds to the fact that blood lactate levels rise much more so than pyruvate during exercise.
Response: The sentence is not correct because LDH converts pyruvate to lactate, an important reaction to re-oxidize NADH, and so reduction of pyruvate to lactate. Lactate during exercise is produced continuously in muscle and released in the blood. Furthermore, it is liver that takes lactate and via gluconeogenesis produces glucose a Cori cycle process where extrahepatic tissues oxidize glucose to lactate (anerobic glycolysis).
Lines 149-151: This sentence is awkward and difficult to read.
Response: The sentences have been revised on line 135 as suggested.
Lines 168-185: This paragraph is somewhat repetitive with information earlier presented in the manuscript.
Response: Sentences have been revised.
Lines 241-261: See comment regards Sections 1.1-1.3.
Response: The questions raised by this reviewer are addressed in our responses to queries for sections 1.1to 1.3.
Figures and Tables: There are way too many figures, a few of them (e.g., figures 1&4, 5 &6) could be combined or are not necessary (e.g. figure 3)
Response: Figures 5 and 6 have been combined as the new Figure 4. One figure was deleted. The figures 1-3 remain although revised. The review has only 5 figures. The revised information on metabolism is necessary as justification and clarification of the metabolic pathways for macronutrients which is the purpose of our review in describing and supporting our novel idea to classifying metabolic and physiologic exerkines.

Reviewer 2 Report
Comments and Suggestions for Authors
the work describes the assigned team extensively, but is monotonous and contributes little to current science. It also has some significant mistakes:
- - the summary is too long
- - chapter 1.1. has no citation
- - lack of purpose of the work in the introduction
- - no separation of materials and methods - it is not described how the literature review was performed
- - Table 1 is missing citation.
- - literature is not prepared as required
Author Response
Reviewer 2 Responses
the work describes the assigned team extensively but is monotonous and contributes little to current science. It also has some significant mistakes:
Response: the use of the word mistakes is vague and should have been used with examples.
- - the summary is too long
Response: We presume the reviewer is referring to the section heading now called 7. Conclusions and Future Perspectives. This section is not long compared with other recent reviews published in Nutrients https://doi.org/10.3390/nu15234919, and 5 paragraphs of conclusion were used in another review article in https://doi.org/10.3390/nu15071729.
- - chapter 1.1. has no citation
Response: Chapter was never used as a heading. Sections were changed; section 2.0 has citations. However, section 2.1 is not a citation of literature but a scientific recommendation on how to classify exerkines which is part of the purpose/objective of the review.
- - lack of purpose of the work in the introduction
Response: Purpose of the review is clearly stated in the abstract lines 57-59 and reiterated in 7. Conclusions and Future perspectives on line 494. The introduction (section 1) is now the headings for the previous heading called exerkines. The purpose and focus are included at lines 81 and 102. Revision was also done at the end of the introduction.
- - no separation of materials and methods - it is not described how the literature review was performed
Response: Our review is not a systematic review, one that is a historical review of exerkine and actions. Our review is a focused evaluation of exerkines associated with metabolism and critical insight into the actions of what we call “metabolic exerkines” involved in energy production for exercise. Our purpose is to present a critical integrative review of exerkines on metabolism.
- - Table 1 is missing citation.
Response: A citation was used in the footnote of Table 1, and the citation # is now added to the footnote. Thank you.
- - literature is not prepared as required
Response: We presume that the reviewer is referring to the font style used in the text which is now changed in the manuscript. For example, Safdar et al. [1] is used in the revision and not Safdar et al. [1]. Thank you.
Round 2
Reviewer 2 Report
Comments and Suggestions for Authors
all suggestions have been corrected by the authors